

# Redressing the balance: quantifying net intercatchment groundwater flows

Laurène Bouaziz[1,2], Albrecht Weerts[2,3], Jaap Schellekens[4], Eric Sprokkereef[5], Jasper Stam[5], Hubert Savenije[1], and Markus Hrachowitz[1]

[1]Water Resources Section, Faculty of Civil Engineering and Geosciences, Delft University of Technology, P.O. Box 5048, NL-2600 GA Delft, The Netherlands
[2]Department Catchment and Urban Hydrology, Deltares, Boussinesqweg 1, 2629 HV Delft, The Netherlands
[3]Hydrology and Quantitative Water Management Group, Department of Environmental Sciences, Wageningen University, The Netherlands
[4]VanderSat, Wilhelminastraat 43A, 2011 VK Haarlem, The Netherlands
[5]Ministry of Infrastructure and Water Management, Zuiderwagenplein 2, 8224 AD Lelystad, The Netherlands

**Correspondence:** Laurène Bouaziz (Laurene.Bouaziz@deltares.nl)

**Abstract.** Intercatchment groundwater flows (IGF), defined as groundwater flows across topographic divides, can occur as regional groundwater flows that bypass headwater streams and only drain into the channel further downstream or directly to the sea. However, groundwater flows can also be diverted to adjacent river basins due to geological features (e.g., faults, dipping beds and highly permeable conduits). Even though intercatchment groundwater flows can be a significant part of the

water balance, they are often not considered in hydrological studies. Yet, assuming this process to be negligible may introduce misrepresentation of the natural system in hydrological models, for example in regions with complex geological features. The presence of limestone formations in France and Belgium potentially further exacerbates the importance of intercatchment groundwater flows, and thus motivates to question the validity of neglecting intercatchment groundwater flows in the Meuse basin. To isolate and quantify the potential relevance of net intercatchment groundwater flows in this study, we propose a three

step approach that relies on the comparison and analysis of (1) observed water balance data within the Budyko framework, (2) results from a suite of different conceptual hydrological models and (3) remote sensing based estimates of actual evaporation. The data of 58 catchments in the Meuse basin provides evidence of the likely presence of significant net intercatchment groundwater flows occurring mainly in small headwater catchments underlain by fractured aquifers. The data suggests that the relative importance of net intercatchment groundwater flows reduces at the scale of the Meuse basin, as regional groundwater

flows are mostly expected to be self-contained in large basins. The analysis further suggests that net intercatchment groundwater flow processes vary over the year and that at the scale of the headwaters, net intercatchment groundwater flows can make up a relatively large proportion of the water balance (on average 10% of mean yearly precipitation) and should be accounted for to prevent overestimating actual evaporation rates.



## 1 Introduction

Water balances rarely close at the catchment scale when assessed with observed data, due to (1) the spatial heterogeneity of precipitation, (2) the low density of the monitoring network especially at high altitudes, (3) the difficulty to measure actual evaporation at the catchment scale, (4) the uncertainty in potential evaporation estimates, (5) errors in precipitation and dis-
charge measurements, and (6) the potential presence of undetected intercatchment groundwater flows (IGF) (Valéry et al., 2010).

Intercatchment groundwater flows are defined as groundwater fluxes crossing topographic divides, implying that precipitation falling in one watershed affects the streamflow in another watershed. A theoretical framework to describe groundwater flows was introduced by Tóth (1963). He classified different systems of groundwater flows, starting from local flow paths,
nested in larger intermediate systems, which in turn are nested in regional flow systems. The theory describes that regional groundwater flow paths transport water from small headwaters to the larger and lower elevation basin, meaning that small basins tend to export or import water and large basins are likely self-contained (Schaller and Fan, 2009). This is based on the assumption that regional flow paths occur within surface drainage boundaries at the largest scale, however, systems with dipping sedimentary beds can divert groundwater away from the basin, leading to complications of the above described theories
and to intercatchment groundwater flows between adjacent basins. (Schaller and Fan, 2009; Frisbee et al., 2016). Regional flow paths within a basin and between adjacent basins are the subject of this study as they characterize intercatchment groundwater flows.

Large scale studies and theoretical models can help to understand the link between intercatchment groundwater flows and physical catchment characteristics. Schaller and Fan (2009) assessed the role of topography, aquifer properties, climate and
geology on intercatchment groundwater flows. On the continental scale, they found that arid climates favor intercatchment groundwater flows. However, on the regional and basin scale, geology exerts the strongest control on intercatchment groundwater flows. The particularities of the geological systems (e.g., faults, connectivity between faults, subsurface flow conduits) can inhibit expected correlations between the magnitude of intercatchment groundwater flows and physical catchment characteristics (e.g. lithology), as was also pointed out by Le Moine et al. (2007). This highlights the difficulty to generalize the
presence of intercatchment groundwater flows based on similarities in climate and topography between watersheds.

Intercatchment groundwater flows cannot be directly measured and are therefore difficult to quantify, which can explain why they are often neglected in small watershed studies (Genereux et al., 2002). However, Schaller and Fan (2009) showed that intercatchment groundwater flows can be a signgicant portion of a basin's water balance across the continental United States; with up to 90% of flow leaving catchments as groundwater export and up to 50% of river flow originating from groundwater
imported from other basins. Methods to identify and quantify intercatchment groundwater flows in real-world basins either rely on stream chemistry and isotope analyses (Genereux et al., 2002; Genereux and Jordan, 2006; Ajami et al., 2011; Frisbee et al., 2011, 2012, 2016), or on water budget analyses (Genereux et al., 2005; Le Moine et al., 2007, 2008; Schaller and Fan, 2009; Hrachowitz et al., 2014). Higher solute concentrations in regional groundwater flows (due to longer residence time) compared to local flow paths can provide evidence for groundwater gains through intercatchment groundwater flows. Water





budget analyses can only show a net gain or loss and not the actual rates of intercatchment groundwater inflow or outflow (Genereux et al., 2002).

Most conceptual hydrological models solely rely on closing the water balance and neglect the possible presence of inter-catchment groundwater flows by relating the change in storage over time to the difference between precipitation and the sum

of actual evaporation and discharge. These models assume watertight catchment boundaries derived from surface elevation, an impermeable substratum and no deep subsurface flow bypassing the stream. These assumptions imply the absence of in-tercatchment groundwater flows. Adding a loss/gain term to represent such intercatchment groundwater flows is often not warranted in models due to limited data availability for calibration (often only streamflow) and the difficulties involved in determining potential and actual evaporation (Beven, 2001). Yet, assuming this process to be negligible may introduce mis-

representation of the natural system in hydrological models, for example in regions with complex geological features (Zhang et al., 2005; Zhang and Savenije, 2005; Reggiani and Rientjes, 2010). Examples of conceptual (or empirical) models that ex-plicitly account for net intercatchment groundwater flows include the GR4J empirical model (Perrin et al., 2003) often applied in French catchments, HYDROLOG (Chiew and McMahon, 1990), SMAR (Goswami et al., 2007; Goswami and O'Connor, 2010), mHM (Samaniego et al., 2011), and the flexible model structure used in Hrachowitz et al. (2014).

Including intercatchment groundwater flows in conceptual models has been studied in a large set of French catchments (Le Moine et al., 2007) and results in a more plausible partitioning between evaporation, streamflow and underground fluxes than methods correcting for potential errors in climatic input data or catchment area instead. Isotopic and chemical analyses indicate an intra-annual variability of intercatchment groundwater flow processes (Ajami et al., 2011; Frisbee et al., 2012).

The overall objective of this study is to detect and quantify net intercatchment groundwater flows (i.e. $Q_{IGF,in} - Q_{IGF,out}$)

in a complementary three step approach through (1) water budget accounting, (2) testing a set of model concepts, and (3) evaluating the results against remote sensing estimates of actual evaporation. In a proof of concept study in the Meuse basin, we test the following hypotheses:

1. Observed water balance data in combination with the Budyko framework can provide robust evidence of the likelihood and spatial variability of net intercatchment groundwater flows.

2. Simple hydrological conceptual models enable to quantify the magnitude and intra-annual variability of net intercatch-ment groundwater flows over meso-scale catchments and to assess the likelihood that intercatchment groundwater flows occur within a basin or between neighboring basins.

3. Actual evaporation estimates from remote sensing provide additional evidence to support the presence of net intercatch-ment groundwater flows.





## 2 Study areas and data

### 2.1 Meuse basin

This study uses data from 58 catchments within the Meuse basin upstream of Eijsden (where the Meuse flows into the Nether-lands), which includes the French and Belgian part of the basin with an area of approximately $21\,000\,\text{km}^2$, see Figure 1. The 58

catchments have areas varying from 50 to $16\,500\,\text{km}^2$, with a median value of $370\,\text{km}^2$ and mean annual precipitation between $750$ and $1200\,\text{mm\,yr}^{-1}$. Median annual runoff and potential evaporation in these catchments is approximately $420\,\text{mm\,yr}^{-1}$ and $620\,\text{mm\,yr}^{-1}$, respectively. Elevation in the basin ranges from 50 to 700 m. The Meuse is a typically rain-dominated river with large variations in seasonal runoff. Snow occurs relatively frequently, but is not a major factor in the discharge regime. The discharge seasonality is mainly caused by high summer and low winter evaporation, as mean precipitation displays little

seasonal variations (de Wit et al., 2001).

The Meuse basin is underlain by a complex geology that combines limestones from the Middle and Late Jurassic in the Southern part of the basin (mainly in the French part) with relatively impermeable metamorphic Cambrian rock and Early Devonian sandstone in the Ardennes Massif and Plateau.

From the 58 available stations, five stations are available in the Semois River catchment (Figure 2 and Table 1) and are

studied in more detail along with five additional stations (Figure 1 and Table 1).

The Semois catchment upstream of Membre-Pont is interesting because it combines both the Jurassic and Early Devonian geological horizons: only the upstream catchment of Sainte-Marie consists of marl (and limestone), while further downstream the basin is underlain by relatively impermeable sandstone and schist. In addition, several discharge stations along the Semois river are available and allow us to detect how net intercatchment groundwater flows ($\text{IGF}_{net}$) evolve as we move further

downstream along the same river. Characteristics of the Semois catchments are included in Table 1 and a map is provided in Figure 2.

In the French part of the Meuse basin, the tributary of the Aroffe River at Vannes-le-Châtel ($198\,\text{km}^2$, see Figure 1) flows underground through limestone deposits towards the Moselle catchment (Fister, 2012). The Aroffe is a typical example of an overflow spring that is activated when the capacity of the conduit is exceeded, while it flows underground to the Moselle the

rest of the time. The Aroffe is one of the additional five catchments where $\text{IGF}_{net}$ are quantified (see Section 4.2.3).

### 2.2 Meteorological and hydrological data

For each catchment, areal averages of precipitation, potential evaporation and observed dicharges are required for the analyses.

Hourly precipitation measurements are interpolated using climatological monthly background grids, using a combination of the HYRAS (Rauthe et al., 2013) and E-OBS (Haylock et al., 2008) datasets and following the method described in van

Osnabrugge et al. (2017). Precipitation measurement in Belgium were provided by the Service Public de Wallonie; in France data was retrieved from the Dutch operational forecasting system. Potential evaporation estimates are based on the Makkink formula (Hooghart and Lablans, 1988) and rely on hourly interpolated temperature station data (using a lapse rate of $6.6\cdot10^{-3}$ °C m$^{-1}$) and hourly radiation data from Maastricht (Royal Netherlands Meteorological Institute). Mean hourly values of



precipitation and potential evaporation are derived from the 1200 m resolution gridded data for each catchment where discharge data is available between 2006 and 2016.

Observed discharge data is available at the hourly time step for the stations in Belgium from the Service Public de Wallonie and at the daily timestep for the stations in France from Banque Hydro. In the Semois catchments, discharges between March

and mid-June 2013 were set to missing because of high observed discharges with too limited precipitation amounts.

### 2.3 Remotely sensed based actual evaporation estimates

Two products of remotely sensed based actual evaporation estimates are used for comparison with modeled actual evaporation.

  – Global Land Evaporation Amsterdam Model (GLEAM, Miralles et al. (2011); Martens et al. (2017))

GLEAM v3a calculates actual evaporation based on satellite-observed soil moisture, vegetation optical depth and snow water

equivalent, reanalysis air temperature and radiation, and a multi-source precipitation product. GLEAM provides actual evaporation estimates at a spatial resolution of $0.25°$ and accounts for subgrid heterogeneity by considering three land surface types (bare soil, short vegetation and vegetation with a tall canopy). GLEAM estimates are available for the entire studied period between 2006-2016.

  – Land Surface Analysis Satellite Applications Facility Daily MSG actual evaporation (LSA SAF, Trigo et al. (2011))

LSA SAF daily MSG (Meteosat Second Generation) actual evaporation (hereafter referred to as LSA SAF) includes soil evaporation, interception and transpiration and is calculated by solving the energy balance by combining radiative, land surface, vegetation and meteorological data. Each pixel (3km·3km resolution at nadir) is split in four tiles to represent main land cover types (bare soil, grassland, crops and forests) and the surface energy balance is solved for each tile separately and results in an actual evaporation value per pixel based on the weighted average of the tiles (https://landsaf.ipma.pt/en/products/

evapotranspiration/dailymet/). LSA SAF estimates are only available for the validation period (2012-2016).

### 3 Methods

This study consists of three parts aimed to identify, quantify and test for the presence of net intercatchment groundwater flows ($\text{IGF}_{net}$) in the Meuse basin. First, we use long term observed water balance data in combination with the Budyko framework (Budyko, 1961) to identify catchments with evidence of water losses or gains through $\text{IGF}_{net}$. Second, we use conceptual

hydrological models to assess the magnitude and temporal variability of potential $\text{IGF}_{net}$ in the Meuse basin and we assume that they are the main cause of water balance discrepancies and thereby neglect uncertainties in forcing data. We model $\text{IGF}_{net}$ as independent losses or gains in alternative model concepts and evaluate their magnitude in several catchments of the Meuse basin. To assess if part of the groundwater flow bypasses the headwater stream to reach the river further downstream, we model the losses/gains in increasingly large catchments along the same tributary. Thirdly, we use actual evaporation from remote

sensing estimates to provide additional evidence for the likelihood and magnitude of $\text{IGF}_{net}$.



### 3.1 Identification: how to detect net intercatchment groundwater flows from observed data signals?

The water balance of a catchment reads:

$$\frac{dS}{dt} = P - Q_{obs} - E_a - Q_{IGF}, \tag{1}$$

where $S$ is the storage in the catchment, $P$ is the precipitation, $Q_{obs}$ is the observed discharge at the catchment outlet, $E_a$ is

the actual evaporation and $Q_{IGF}$ is the groundwater net loss (if $Q_{IGF}$ is positive, meaning that the groundwater flow out of the catchment is larger than the flow into the catchment) or net gain (if $Q_{IGF}$ is negative) to the catchment.

Intercatchment groundwater flows are often not considered and over a long period (several years), the change in storage is assumed to be zero, which reduces the equation to:

$$P = Q_{obs} + E_a, \tag{2}$$

The Budyko framework (Budyko, 1961) describes the empirical global relation between the long term evaporative index ($E_a/P$) and the dryness index ($E_p/P$, with $E_p$ the potential evaporation) and shows that natural catchments show a tendency to plot along the Budyko curve in the theoretical range located in between the energy and water limits. The water limit implies that a catchment cannot evaporate (or discharge) more water than it receives from precipitation, this implies that catchments with higher runoff than precipitation plot beyond the water limit (gaining catchments) in the Budyko framework. The energy

limit implies that catchments cannot evaporate ($E_a$) more than the energy available for evaporation ($E_p$), therefore catchments where the difference between precipitation and runoff is larger than potential evaporation are beyond the energy limit (leaky catchments), as shown in Figure 3. Assuming negligible observation errors, they are likely affected by net intercatchment groundwater inflows (gaining catchments) or outflows (leaky catchments). Andréassian and Perrin (2012) suggest to replace the axis of the evaporative index ($E_a/P = 1 - Q_{obs}/P$) with the runoff coefficient ($Q_{obs}/P$) in the Budyko framework because

gaining catchments would otherwise have a negative evaporative index and because $E_a$ itself is not measured at the catchment scale. We therefore plot each catchment in the non-dimensional representation of the runoff coefficient ($Q_{obs}/P$) as a function of the dryness index ($E_p/P$), hereinafter referred to as the Budyko framework for the sake of convenience, using hydrological years between October 2006 and September 2016 (10 years) with more than 350 days of streamflow data per year.

Catchments show a tendency to plot close to the Budyko curve or other alternative expressions. The Turc-Pike formula (Turc,

1954; Pike, 1964) plots very close to the Budyko curve (Figure 3) and has often been used in studies of French catchments (Le Moine et al., 2007) and was therefore applied in our analysis. The Turc-Pike formula reads:

$$\frac{Q_{obs}}{P} = 1 - \frac{1}{(1 + (\frac{P}{\alpha \cdot E_p})^{\gamma})^{\frac{1}{\gamma}}}, \tag{3}$$

with parameters $\alpha = 1$ and $\gamma = 2$. Depending on the values of the parameters $\alpha$ and $\gamma$, the Turc-Pike relation can span the entire domain from the energy limit to the water limit. Here we define catchments plotting more than 5% away from this curve

(which implies a narrower range than in Gentine et al. (2012) but wider than in Li et al. (2014)) and close to the limits as likely to be affected by IGF$_{net}$. More specifically, catchments plotting beyond the energy limit and between the energy limit





and the lower boundary of the Turc-Pike uncertainty range (see Figure 3) potentially indicate the presence of net subsurface losses. Indeed, catchments that plot very close to the energy limit imply that the difference between precipitation and discharge approximates the total energy available for evaporation ($P - Q_{obs} \approx E_p$). During dry and/or very warm periods, however, evaporation is constrained by water availability and mean yearly actual evaporation is therefore expected to be considerably

lower than potential; this in turn means that water must be leaving the catchment through another route to comply with the observed long term water balance. We hypothesize that water is leaving the catchment through underground pathways.

We consider the shortest distance between each catchment and the energy limit in the Budyko framework as a proxy for the presence of IGF$_{net}$. The closer a catchment is to the energy limit, the higher the probability of IGF$_{net}$. We adjust this distance by the shortest distance of the point on the Turc-Pike curve at the catchment's $E_p/P$ to the energy limit (see Figure 3) because

arid catchments have lower runoff coefficients and are therefore expected to be futher away from the energy limit. Negative distances imply that catchments plot beyond the energy limit.

We then assess if the adjusted distance to the energy limit is correlated with several physical catchment characteristics that may influence the formation of IGF$_{net}$, including the percentage of highly productive fissured aquifers (including karstified rocks) as provided by the International Hydrogeological Map of Europe (IHME, www.bgr.bund.de/ihme1500) / International

Geological Map of Europe (IGME), catchment area and percentage of hillslopes (slopes steeper than 13%, Gharari et al. (2011)).

### 3.2 Quantification: how to quantify the variation of net intercatchment groundwater flow processes over the Meuse basin using conceptual models?

#### 3.2.1 Models description

A reference conceptual model is developed including interception, soil moisture, fast and slow reservoirs, but no IGF$_{net}$ (see Figure 4). This conceptual model has ten calibration parameters. The characteristic time scale of the recession of the slow reservoir is determined with a master recession curve analysis.

Two options are investigated to incorporate IGF$_{net}$ in the reference model. The first one involves a continuous constant groundwater exchange flux (loss/gain) from/to the slow reservoir ($Q_{IGF}(t) = C_{IGF}$), assuming a slowly draining, homoge-

neous, low-permeability aquifer. The second relies on preferential permeable pathways, activated above a certain threshold, to lose or gain water, (see Figure 4 and Section 1 and 2 of the Supplement). In the preferential model, part of the recharge is lost or gained (before entering the slow reservoir) when the recharge exceeds a certain threshold. An error function is used to simulate this behaviour: $Q_{IGF}(t) = erf(R_{US}(t), \mu, m_3) \cdot P_{erc} \cdot R_{US}(t)$, with $R_{US}(t)$ the recharge from the root zone storage to the slow reservoir, $\mu$ the threshold parameter of the recharge above which IGF$_{net}$ occurs, $P_{erc}$ the maximum fraction of the

recharge to IGF$_{net}$ and $m_3$ a shape parameter of the error function (not calibrated). The constant loss/gain model resembles the one in Hrachowitz et al. (2014) and counts one extra parameter, while the preferential IGF$_{net}$ model has two additional parameters compared to the reference model.





In the catchment of the Aroffe River, water sinks in the karstified limestone after travelling through sandstone and marl deposits and emerges again in the neighboring catchment of the Moselle (which is a tributary of the Rhine River). During peak flows, the conduit capacity is exceeded and water flows in the river bed of the Aroffe (Fister, 2012). To simulate the hydrological functioning of the Aroffe river, an overflow type of model is developed to quantify the losses of this catchment

to the neighboring Moselle basin, according to $Q_{IGF}(t) = K_{IGF}^{-1} \cdot S_S(t)$, with $K_{IGF}$ the characteristic time scale of the underground stores ($S_S$), as shown in Figure 4 and in Section 1 of the Supplement.

Parameters, water balance and constitutive equations of all models are provided in Section 1 and 2 of the Supplement and model schematizations are shown in Figure 4. All models are programmed in Python and an implicit Euler time stepping scheme is used to solve the model equations.

### 3.2.2  Model experiments - general procedure

The model was run between 1 January 2006 to 31 December 2011, using 2006 as a warm-up year, to explore the parameter space with a Monte-Carlo strategy, sampling from uniformed prior parameter distributions ($10^5$ realizations). This was done at an hourly time step because of the fast processes occurring in the Meuse River basin. Feasible parameter sets are retained based on their simultaneous ability to reproduce high and low flow metrics during calibration with Nash-Sutcliffe efficiencies

of at least 0.7 for different indicators (Nash-Sutcliffe efficiency of the flows $E_{NS,Q}$ and of the log of the flows $E_{NS,logQ}$, Nash-Sutcliffe efficiency of the flow duration curve of the log of the flows $E_{NS,FDC,logQ}$), and to reproduce discharge volumes at different temporal scales (relative volume error $E_{RVE}$, Nash-Sutcliffe efficiency of runoff coefficients for 6-monthly $E_{RC,6m}$, monthly $E_{RC,m}$ and weekly $E_{RC,w}$ periods). The tested models are evaluated in an independent validation period running from 1 January 2012 to 31 December 2016.

Prior and posterior parameter ranges are provided in Section 2 of the Supplement. The characteristic time scale of the recession of the slow reservoir is estimated with a master recession curve analysis for each catchment (Fenicia et al., 2006). A range of 10 days around the derived value is used as a calibration range to account for non-linear recession when a constant loss/gain is added to the slow reservoir.

The experiments designed to test the hypotheses of this paper are described in the following Sections.

### 3.2.3  Representation: how to represent net intercatchment groundwater flows: zero, constant or preferential flows?

The stations on the Semois River and its tributary (Vierre at Straimont and Semois at Sainte-Marie, Tintigny, Chiny, Membre-Pont shown in Figure 2) are used to assess three alternative model concepts: the reference model without IGF$_{net}$, constant IGF$_{net}$ from/to the slow reservoir and preferential IGF$_{net}$ from/to the recharge to the slow reservoir. These stations are selected because they also allow us to quantify how IGF$_{net}$ evolve from upstream to downstream along the same river (see

Section 3.2.4). The five stations are calibrated independently using the three models to quantify the magnitude of IGF$_{net}$ in the subsequent catchments. The most suitable model structure is determined based on a visual inspection of hydrographs and modeled discharge regime, a comparison of performance indicators in the validation period, and a comparison between the magnitude of the loss and the distance to the energy limit (long term mean and annual variability). Additionally, modeled mean





annual actual evaporation are compared to Turc-Pike estimates and we assess the shift of the modeled water balance in the Budyko framework when $IGF_{net}$ are considered versus neglected.

### 3.2.4 Direction: where do intercatchment groundwater flows go?

To test if part of the groundwater flow bypasses the headwater stream to reach the river only further downstream, we model the Semois River catchments (using the experiments described in Section 3.2.3) to quantify how the loss/gain term varies as catchment size increases along the same river. Additionally, we looked for examples in the literature located in the Meuse basin to highlight the possible difference between $IGF_{net}$ that are internal to a river basin and $IGF_{net}$ to neighboring river basins.

### 3.2.5 Magnitude: what is the magnitude of net intercatchment groundwater flows at the scale of the Meuse basin?

Several catchments plotting close or beyond the energy limit (from the analysis described in Section 3.1) are modeled to quantify the magnitude of potential $IGF_{net}$ at several locations in the Meuse basin. Additional catchments where the magnitude of $IGF_{net}$ are evaluated using the preferential model (because it performed better for the Semois at Sainte-Marie, see the results in Section 4.2.1) include the Sormonne at Belval, the Mehaigne at Huccorgne, the Bocq at Yvoir and the Crusnes at Pierrepont (Figure 1). For the Aroffe at Vannes-le-Châtel, the overflow type of model (Figure 4) is used to model the loss towards the Moselle basin, based on findings from literature (Fister, 2012).

### 3.3 Evaluation: is the presence of net intercatchment groundwater flows supported by remotely sensed actual evaporation estimates?

We test for the presence of $IGF_{net}$ using independent additional data sources. Actual evaporation is a major component of the water balance at the catchment scale, but it is also a great unknown. Reliable estimates of actual evaporation at the catchment scale would allow us to attribute the gap in the water balance to $IGF_{net}$, assuming minor anthropogenic activities. Global evaporation products are however not derived directly from earth observations, but rely on remotely sensed data in combination with models to derive actual evaporation. In this study, we compare two sources of remotely sensed actual evaporation estimates (LSA SAF and GLEAM) with our modeled actual evaporation to test the hypothesis of $IGF_{net}$.

## 4   Results and discussion

### 4.1   Identification: observed data and Budyko framework to detect net intercatchment groundwater flows and link with physical catchment characteristics

The analysis of observed water balances in the Budyko framework shows that relatively small headwater catchments of the Meuse basin (50-700 km$^2$, see Figure 1) plot closest to or beyond the energy limit (Figure 3), this suggests that these catchments exhibit the highest potential for the presence of net intercatchment groundwater flows ($IGF_{net}$). Amongst them is the headwater catchment of the Semois at Sainte-Marie (Figure 2) which plots close to the energy limit, suggesting underground losses



towards other catchments. The water balance of two catchments in the North-East (Figure 1) might be affected by the presence of dams (FAO, 2016) and the two catchments are therefore left out of further analyses. The net losses calculated with long term observed runoff, precipitation and Turc-Pike estimates of actual evaporation in these headwater catchments range between 70 mm y$^{-1}$ (for the Semois at Sainte-Marie, which corresponds to 7% of mean annual precipitation) to 260 mm y$^{-1}$ (for

the Aroffe catchment at Vannes-le-Châtel, which is 31% of annual precipitation), with a median of 100 mm y$^{-1}$ (or 12% of median annual precipitation). The distance of the Aroffe catchment to the energy limit is negative (the catchment plots beyond the energy limit) and approximately three times larger than the (positive) distance of the Semois at Sainte-Marie.

The catchments of the Meuse basin show a significant trend (p=0.001 and R$^2$=0.22) indicating more losses from the catchment (negative or shorter distance to the energy limit) as the percentage of highly productive fissured aquifers increases, as

shown in Figure 5. Intercatchment groundwater flows in the Meuse basin are therefore likely to occur in catchments with highly productive fissured aquifers, including karstified rocks (see the IHME hydrogeological map in Figure 1). These productive aquifers are characterized by limestone, marl or chalk lithologies (IGME). Karstification processes may cause 'piracy' routes to develop (Hartmann et al., 2014) and therefore be at the origin of IGF$_{net}$.

We use the percentage of hillslopes in a catchment, (defined as areas with a slope steeper than 13%, Gharari et al. (2011)) as

a proxy for how well the drainage network is defined from the surface and relate it to the potential presence of IGF$_{net}$ (through the distance to the energy limit) as shown in Figure 5. The data shows a significant trend (p=0.001 and R$^2$=0.22) indicating less losses from the catchment (larger distance towards the energy limit) as the percentage hillslope increases. The underlying idea is that surface topography displays the result of a competition between surface and subsurface flows. Catchments dominated by steep valleys, as encountered in the Ardennes, clearly show their drainage network at the surface. The steeper, the higher

the relative importance of lateral flow through a subsurface preferential path network to the channel/stream. On the other hand, catchments lying on permeable lithologies as chalk and middle Jurassic limestones may be dominated by rivers cutting through relatively flat plateaus and may hide an underground network of subsurface flow paths from the surface (Le Moine, 2008). The flatter, the higher the potential importance of an underground flow network and therefore of subsurface losses/gains. In the Meuse basin, IGF$_{net}$ are therefore likely to occur in catchments dominated by a relatively flat topography.

We also tested the hypothesis that part of the groundwater flow bypasses the channel to reach the river only further downstream by correlating the distance to the energy limit (as a proxy for the presence of IGF$_{net}$) to catchment area for the main tributaries of the Meuse basin (Figure 5). We expected the presence of IGF$_{net}$ to be reduced as catchment size increases, and although this trend is significant (p=0.032 and R$^2$=0.10), the correlation is weak. The data shown in Figure 5, however, suggests that evidence for IGF$_{net}$ is highest in small catchments (with areas less than 500 km$^2$) and much less pronounced in

larger downstream catchments, although there are also small catchments with little evidence for it. This is likely related to the variability of local geological features underlying these small catchments.





## 4.2 Quantification: variation of net intercatchment groundwater flow processes across the Meuse basin

### 4.2.1 Representation: a preferential model to represent net intercatchment groundwater flows

The reference (without $IGF_{net}$), constant and preferential $IGF_{net}$ models are calibrated on subsequent catchments along the Semois river. In the following sections, the models are evaluated based on (1) performance indicators during the validation period and visual inspection of the hydrographs and seasonal behavior, (2) the magnitude of modeled $IGF_{net}$, and (3) modeled actual evaporation.

– Performance indicators and visual inspection of the hydrographs

Performance indicators of the feasible realizations of the three models in the Semois catchments during the calibration and validation period are shown in Figure 6. The preferential model shows an improvement in high and low flow indicators, and in modeled runoff coefficients in the Semois catchment at Sainte-Marie compared to the constant and zero $IGF_{net}$ models; whereas in the other catchments of the Semois River, performance indicators are similar for the three models. Nash-Sutcliffe efficiencies of daily flows ($E_{NS,Q}$) and log of the flows ($E_{NS,logQ}$) increase when the reference model (no $IGF_{net}$) is extended with a constant $IGF_{net}$ term and increase even more when a preferential $IGF_{net}$ term is included in the catchment upstream of Sainte-Marie. This also applies for the Nash-Sutcliffe efficiency applied on monthly and weekly runoff coefficients ($E_{NS,RC,m}$ and $E_{NS,RC,w}$). On the other hand, all performance indicators for the Vierre at Straimont (Figure 6) show similar results for the three models. Adding an exchange term in this sandstone dominated catchment (constant or preferential) does not lead to an improved performance. This behaviour also characterizes the other catchments at Tintigny, Chiny and Membre-Pont (Figure 6).

A visual inspection of the in 2014 modeled and observed hydrographs at Sainte-Marie (Figure 7) shows a decrease in modeled winter peak flows at the beginning of the year and an increase of modeled peak flows after the dry season (October) for the preferential model compared to the zero $IGF_{net}$ model, which better approximates observed behavior. Although this behavior might vary throughout the years, a higher performance of the preferential model in reproducing the observed discharge regime is also visible in Figure 8. Including preferential $IGF_{net}$ in the model reduces the mean overestimation of 9 mm month$^{-1}$ at the beginning of the year and the underestimation of 11 mm month$^{-1}$ in October and November simulated by the zero $IGF_{net}$ model to respectively 0.5 and 3 mm month$^{-1}$ on average. This implies that the error is reduced by 94% at the beginning of the year and by 73% in October and November. The improved simulation of the seasonal behavior indicates a better representation of the underlying processes and the resulting partitioning of water fluxes.

An analysis of the inter-annual variability of modeled $IGF_{net}$ (see Section 3 of the Supplement) also shows better performances achieved with the preferential $IGF_{net}$ model.

– Groundwater net loss/gains in the Semois catchment

In the catchment upstream of Sainte-Marie, a median annual loss term of 17% and 20% of observed discharge (corresponding to 77 and 90 mm y$^{-1}$) is modeled by the feasible realizations of the preferential and constant $IGF_{net}$ model, respectively, as



shown in Figure 9. The magnitude of $\text{IGF}_{net}$ decreases in the catchments further downstream on the Semois River. At the catchment outlet (Membre-Pont) and in the Vierre tributary, the magnitude of $\text{IGF}_{net}$ is centered around zero. The range of $\text{IGF}_{net}$ is larger for the constant model compared to the preferential model. For the preferential model, $\text{IGF}_{net}$ approximate a value of zero for all other catchments than Sainte-Marie. In the constant model, median values of $\text{IGF}_{net}$ are positive (losses),

but some realizations imply a slight gain. Additionally, Figure 9 shows that the magnitude of $\text{IGF}_{net}$ decreases as the distance to the energy limit increases. This means that as catchments plot closer to the Budyko curve (and further away from the energy limit), we see the relative importance of $\text{IGF}_{net}$ decreasing, which is in line with expectations.

 – Effect on actual evaporation

Turc-Pike estimates of actual evaporation are compared with modeled mean yearly actual evaporation of the feasible realiza-
tions of the three models in all Semois stations in Figure 10. Including (constant or preferential) $\text{IGF}_{net}$ in the catchment of Sainte-Marie leads to median annual actual evaporation rates close to Turc-Pike estimates; whereas the reference model leads to 10% higher actual evaporation rates (535 mm yr$^{-1}$ for the preferential model versus 590 mm yr$^{-1}$ for the zero $\text{IGF}_{net}$ model). The reference model compensates for the absence of an intercatchment groundwater flow term by increasing actual evaporation rates to reproduce observed flow volumes. For the majority of the other catchments, the effect of adding $\text{IGF}_{net}$
on modeled actual evaporation rates is less pronounced, but still visible.

 When using observed river discharges and neglecting $\text{IGF}_{net}$, the catchment of Sainte-Marie plots close to the energy limit in the Budyko framework; however, when $\text{IGF}_{net}$ are modeled and added to river flows, the catchment of Sainte-Marie plots close to the Turc-Pike curve, as shown in Figure 11. This shift in the Budyko framework occurs because we acknowledge that part of the produced discharge from the catchment bypasses the measuring gauge in the river. Including $\text{IGF}_{net}$ in the representation
of the system results in a higher degree of plausibility, based on the Budyko framework. The shift is most obvious for the catchment of Sainte-Marie, although it also occurs in the other catchments.

**4.2.2  Direction: groundwater bypass routes versus intercatchment groundwater flows to external basins**

The magnitude of modeled $\text{IGF}_{net}$ decreases from nested upstream to downstream catchments along the Semois River (see Figure 9), which is an indication that 'losses' modeled at Sainte-Marie are internal to the catchment of the Semois River. Losses
in the upstream catchment of Sainte-Marie reappear as additional groundwater inflows in the downstream parts of the Semois, thereby reducing the $\text{IGF}_{net}$ from upstream to downstream.

 In contrast, experiments previously conducted in the Aroffe River catchment (Fister, 2012; Martin and Zany, a) revealed the presence of groundwater flows leaving the Meuse basin towards the Moselle catchment (which is part of the Rhine basin). Losses from the Meuse basin also occur along the northern boundary of the tributary of the Geer River catchment (Reggiani
and Rientjes, 2010). Additionally, downstream of the village of Bazoilles, the Meuse flows underground during a large part of the year, leaving its surface bed empty, before emerging again at Noncourt, just upstream of Neufchâteau (in the upstream part of the Meuse basin); this is referred to as 'les pertes de la Meuse' (Newman, 1949; Martin and Zany, b). This variety of





processes highlights the contrast between stations that are losing water to neighboring catchments (Aroffe to the Rhine) and catchments that are losing water to themselves further downstream.

### 4.2.3 Magnitude: quantification of net intercatchment groundwater flows at the scale of the Meuse basin

The magnitude of $IGF_{net}$ is assessed in several other catchments of the Meuse basin that plot close or beyond the energy limit
(annotated catchments in Figure 1). The preferential model is used to assess the magnitude of $IGF_{net}$ because it performed better in the catchment of Sainte-Marie. In the Aroffe catchment, an overflow type of model is applied to represent the functioning of the system based on a priori available knowledge (see Section 3.2.1). Modeled mean annual flows between 2007 and 2016 overlap well with observations as shown in Figure 12. The ratio of mean annual net intercatchment groundwater flows over observed discharges is always positive (indicating a loss). Modeled losses can be substantial compared to observed discharges
as shown in Figure 12. In the Aroffe, the median loss rate (of 208 mm y$^{-1}$) is approximately 2.5 times higher than observed river flows (85 mm y$^{-1}$). Median values of yearly loss rates over observed discharges range from 0.1% to 32% (0.3 to 130 mm yr$^{-1}$) in the other catchments. Modeled actual evaporation is close or slightly overestimates Turc-Pike estimates (Figure 12), showing that the models are able to reproduce the observed long term water balance in a meaningful way.

At the scale of the Meuse basin, intercatchment groundwater flow processes play a little role because they occur in relatively
small catchments and because part of these losses may be internal to the Meuse basin. However, $IGF_{net}$ occurring at the scale of headwater catchments make up a considerable part of the water balance (on average 10% and up to 25% of mean annual precipitation), which in many current models is wrongly attributed to actual evaporation.

### 4.3 Evaluation: comparison with actual evaporation from remote sensing

GLEAM estimates of mean yearly actual evaporation approximate or slightly overestimate (< 5%) modeled and Turc-Pike
estimates of actual evaporation, as shown in Figure 10 and 12, whereas estimates from a land surface modeling approach, as LSA SAF data are considerably lower (between 400 and 470 mm yr$^{-1}$, Figure 12) in the studied catchments. While the difference in both products highlights the uncertainty in remotely sensed based estimates of actual evaporation, it also shows that actual evaporation might even be less than resulting from our models, which might imply even larger magnitudes of losses due to $IGF_{net}$. The simple conceptualization of soil moisture constrained evaporation used in our models, which does
not account for a temperature based stress function, might lead to an overestimation of transpiration. Thus, being arguably conservative modeled estimates, the low estimates of LSA SAF evaporation lend further credibility to evidence suggesting the presence of considerable $IGF_{net}$.



## 5  Limitations and advances

### 5.1  Limitations

In this work, we rely on the empirical organizing principle provided by the Turc-Pike or Budyko curves (Turc, 1954; Budyko, 1961; Pike, 1964) and assume that catchments of the Meuse basin plotting close or beyond the energy limit (Figure 3) may

be subject to losses due to net intercatchment groundwater flows. Changing vegetation, climate and human interactions might, however, also be at the origin of catchments deviating from the Budyko curve (Velde et al., 2014; Berghuijs et al., 2014). The location of each catchment within the Budyko framework is also subject to uncertainties in the data used to calculate long term averages of precipitation, discharge and potential evaporation. Data uncertainties can originate from the spatial interpolation of the precipitation, the choice of a potential evaporation formula, errors in discharge measurements or in catchment delin-

eation, or the presence of unknown anthropogenic activities affecting the water balance. In spite of these shortcomings, the three step approach of this study, which combines different perspectives and data to estimate net intercatchment groundwater flows, allowed us to plausibly attribute deficits in the observed water balance to the potential presence of net intercatchment groundwater flows.

We treated intercatchment groundwater flows as independent net losses or gains in lumped conceptual catchment models,

without explicitly connecting the loss of one catchment to the gain of another. By modeling several stations along a same tributary (the Semois), we hypothesized that the loss in the headwater catchment at Sainte-Marie might bypass the channel to reach the river only further downstream, implying an 'internal' loss within the river system; but other configurations of groundwater flows in this area might lead to similar results. Additionally, we found evidence in literature (Fister, 2012) that the Aroffe catchment flows underground to the Moselle catchment (a tributary of the Rhine), but we could not relate the flow out of

the relatively small Aroffe catchment (198 km$^2$) to its emergence in the much larger Moselle catchment near Toul (3338 km$^2$) due to the difference in catchment area. Interestingly, in a recent geological past (250,000 years ago), the upstream catchment of the Moselle at Toul was flowing through the Meuse valley before it changed course to join the Rhine basin (de Wit, 2008). Subsurface flow paths connecting both catchments may therefore still remain from these earlier geological times.

### 5.2  Advances

In this study, we question in three steps the validity of neglecting intercatchment groundwater flows in catchment-scale hydrological studies. In the Meuse basin, the potential presence of net intercatchment groundwater flows is detected from observed water balance data in relatively small headwater catchments ($< 500$km$^2$) and is much less pronounced in larger downstream catchments (Figure 1). In the theory advanced by Tóth (1963), regional groundwater flows occur from the headwaters to the bottom of the basin. This implies that headwater catchments may export water through groundwater flow paths into the river

further downstream, thereby increasing the groundwater contributions in larger downstream catchments; this suggests a variability of dominant hydrological (subsurface) processes across spatial scales, as also demonstrated by Frisbee et al. (2011). Schaller and Fan (2009) found that the largest magnitudes of intercatchment groundwater flow occur at catchment size near 100 km$^2$, which also results from our analysis (Figure 5). They also report that efficient aquifers favor intercatchment ground-



water flows. In the Meuse basin, the identified headwater catchments are underlain by highly productive and fissured aquifers, where karstification processes might be at the origin of underground exchange flow paths between catchments. The relatively weak correlations between physical catchment characteristics and intercatchment groundwater flows shown in Figure 5 can be explained by the high spatial variability of intercatchment groundwater flows due to local geological features that overrule

theoretical relations at the basin scale, as also argued by Genereux et al. (2002); Schaller and Fan (2009); Frisbee et al. (2016).

We make a first step to bridge the gap between regional groundwater models where topographic catchment boundaries are not considered and lumped conceptual hydrological models that treat catchments as well-defined impermeable entities, by adding an additional flux in conceptual models to represent net intercatchment groundwater flows. We model net inter-catchment groundwater flows as preferential fluxes, occurring when recharge exceeds a threshold, to represent the filling of

underground stores before intercatchment flows paths are activated, rather than as constant matrix flow. Interestingly, we show that accounting for net preferential intercatchment groundwater flows not only improves low flow performance indicators, but also high flow simulations. The increased performance achieved with the preferential model during both high and low flows suggests that streamflow generation processes, and especially the relative importance of intercatchment groundwater flows change throughout the year, as also found by Frisbee et al. (2012) based on a chemical and isotopic analysis. Ajami et al.

(2011) also suggest that local, intermediate and regional groundwater flow paths are active during winter, while mainly local groundwater flow paths are active during summer. The ratio of net intercatchment groundwater exports over total discharge ($Q_{IGF}/(Q_{IGF}+Q_{river})$) is about 70% in the Aroffe catchment (where the flow is diverted into the neighboring Moselle river) and is on average 17% in the other catchments, these values are within the range provided by Schaller and Fan (2009).

We use independent data sources of remotely sensed actual evaporation estimates to quantify the overestimation of actual

evaporation modeled when intercatchment groundwater flows are neglected. Both global actual evaporation products (GLEAM and LSA SAF) rely on different models and remotely sensed data and provide relatively large differences in mean yearly values (up to 150 mm year$^{-1}$), highlighting the large uncertainty in estimating actual evaporation. While GLEAM actual evaporation estimates approximate our model results and Turc-Pike estimates, LSA SAF estimates indicate lower evaporation rates, potentially indicating an underestimation of actual evaporation in this area, or the even larger importance of losses due

to net intercatchment groundwater flows in the studied catchments.

## 6   Conclusions

This proof of concept study in the Meuse basin shows strong evidence that we can identify net intercatchment groundwater flow processes from analyzing the long term observed water balance of a catchment. The results suggest that intercatchment groundwater flows mainly play a role in headwater catchments ($< 500$ km$^2$) with productive aquifers. In these catchments,

we then use simple conceptual models to show that a net groundwater loss occurs when recharge exceeds a threshold. This preferential net loss term represents the filling of underground stores before intercatchment flow paths are activated, and ranges between 0 and 208 mm yr$^{-1}$ (0 and 25% of annual precipitation) with an average of 100 mm yr$^{-1}$ (10% of mean annual precipitation) in the studied catchments. Some of these underground flow paths may lead to downstream catchments along the





same river (regional groundwater flow paths), while others may lead to neighboring river basins (diverted groundwater flows due to the presence of geological features), which explains why these net losses can be considerable at the headwater catchment scale and negligible at the scale of larger basins (modeled net intercatchment groundwater flows reduced to zero at the most downstream station of the Semois tributary). These findings therefore highlight that dominant streamflow generation processes

vary across spatial scales. Additionally, errors in simulating the seasonal behavior are reduced by more than 70% with the preferential model, this suggests a pronounced intra-annual variability of the magnitude of net intercatchment groundwater flow processes. Neglecting net intercatchment groundwater flows in conceptual models may still result in high performances of streamflow simulation, however, it comes at the cost of overestimating actual evaporation rates to compensate for this lack. Including net intercatchment groundwater flow processes in models can considerably increase the correspondence between

modeled actual evaporation and remote sensing estimates, this provides additional evidence for the presence and magnitude of net intercatchment groundwater flows.

*Competing interests.* The authors declare that they have no conflict of interest.

*Acknowledgements.* The authors would like to thank the Service Public de Wallonie, Direction générale opérationnelle de la Mobilité et des Voies hydrauliques, Département des Etudes et de l'Appui à la Gestion, Direction de la Gestion hydrologique intégrée (Bld du Nord 8-5000

Namur, Belgium) for providing the precipitation and discharge data.





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



**Figure 1.** Right: Digital elevation model and outline of the Meuse basin with all catchments (black), catchments plotting beyond the energy limit (red), catchments very close to the energy limit (orange). The location of the Semois catchment at Membre-Pont is indicated in pale turquoise. Right: International Hydrogeological Map of Europe (IHME), location of main dams (black squares, FAO database) and catchments close to (orange) and beyond (red) the energy limit.





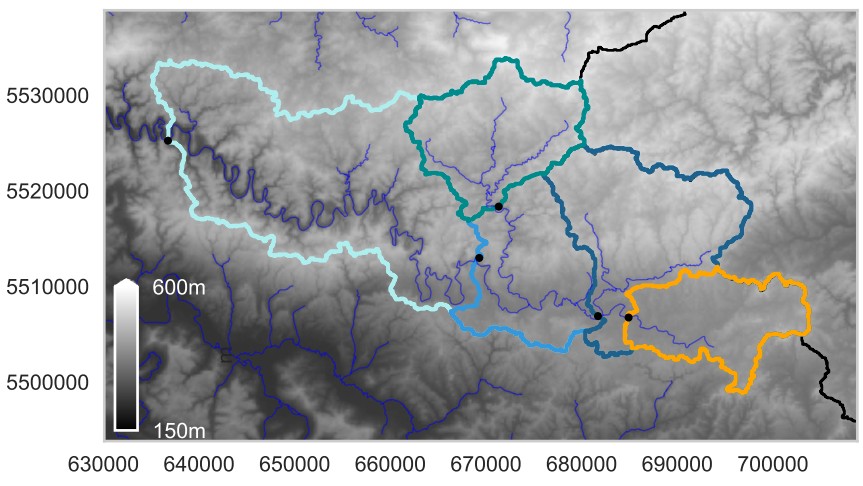

**Figure 2.** Digital elevation model of the Semois catchment and location of the stations from upstream to downstream: Sainte-Marie (orange), Tintigny (dark blue), Chiny (light blue), Membre-Pont (pale turquoise) and additional tributary in the North is the Vierre at Straimont (blue-green). The catchment of Sainte-Marie plots very close to the energy limit as shown in Figure 3.





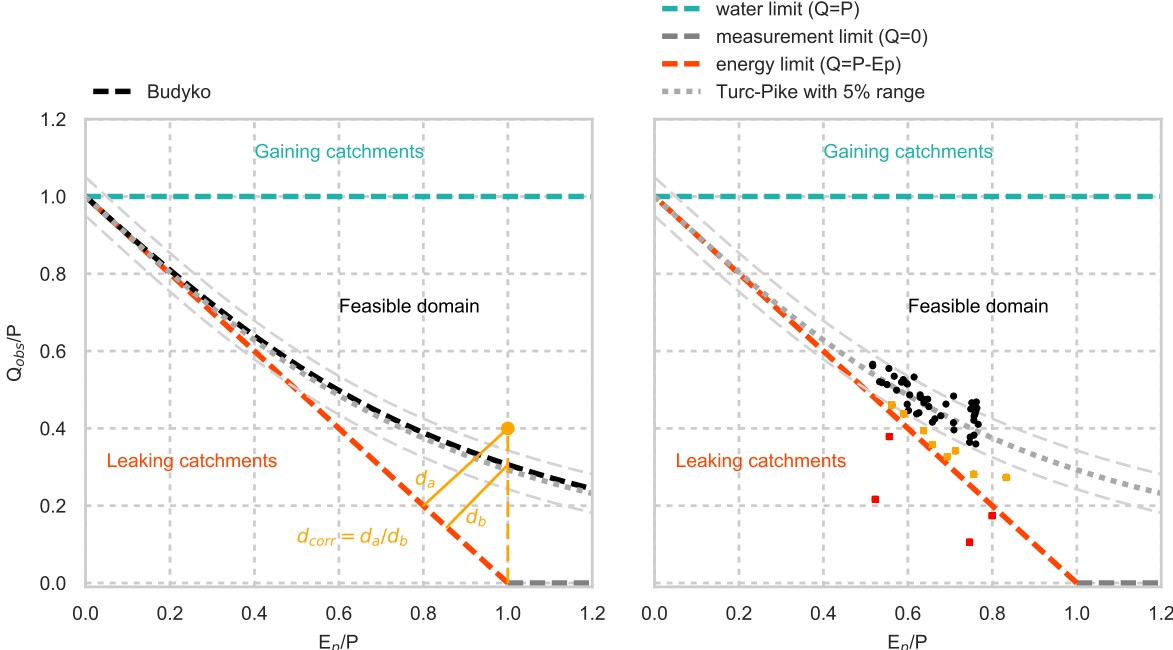

**Figure 3.** Left: Dimensionless representation of the runoff coefficient ($Q_{obs}/P$) as a function of the dryness index ($E_p/P$), referred to as the Budyko framework. The red line is the energy limit ($Q_{obs} = P - E_p$) beyond which catchments are leaking water; the blue line is the water limit ($Q = P$) above which catchments are gaining water; the dark grey line is the measurement limit ($Q = 0$). The domain within these three limits is the theoretical feasible domain. The Turc-Pike and the Budyko curves plot very close to each other. The 5% uncertainty bound around the Turc-Pike curve is also shown. For each catchment, the ratio of the distance to the energy limit ($d_a$) over the distance of Turc-Pike to the energy limit ($d_b$) is used as a proxy for the presence of net intercatchment groundwater flows. Right: the catchments of the Meuse basin are located around the Turc-Pike line (black circles). However, four catchments plot beyond the energy limit (red squares) and eight catchments plot very close to the energy limit and are beyond the lower 5% range of Turc-Pike (orange squares). In these catchments, we expect net intercatchment groundwater flow losses to occur.



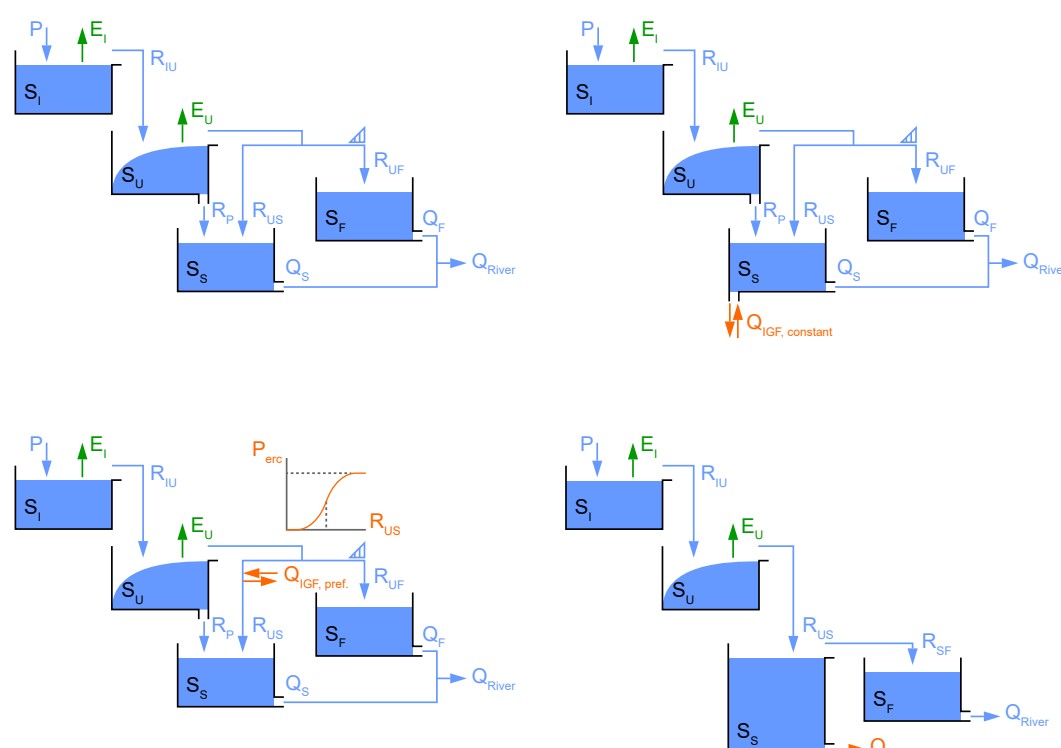

**Figure 4.** Conceptual model schematizations. Upper left: reference model without net intercatchment groundwater flows. Upper right: reference model with net constant intercatchment groundwater flows from the slow reservoir. Lower left: reference model with net preferential intercatchment groundwater flows retrieved from or added to the recharge to the slow reservoir (using an error function that relates the percentage or recharge lost or gained to net intercatchment groundwater flows as a function of the recharge rate). Lower right: overflow model used for the Aroffe catchment at Vannes-le-Châtel that simulates river flows in the Aroffe only when the underground storage capacity is exceeded. The rest of the time, flows occur underground towards neigboring basins. Here, we define $P$ as precipitation, $E$ as evaporation, $S$ as storage, $R$ as an internal flux and $Q$ as surface or subsurface discharge. For the subscripts, we define $I$ as interception, $U$ as root zone, $S$ as slow response, $F$ as fast response, $P$ as percolation. The parameter $P_{erc}$ defines the maximum percentage of recharge as net intercatchment groundwater flow.



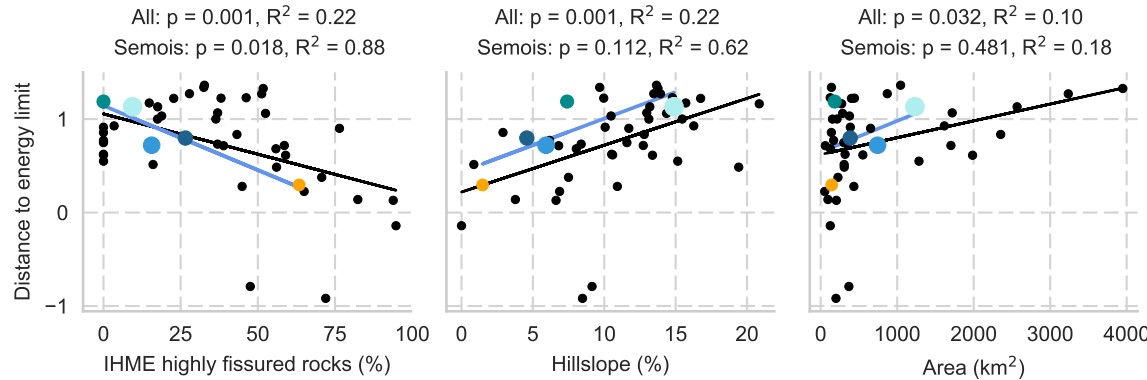

**Figure 5.** Ajusted distance of each catchment to the energy limit in the Budyko framework (as explained in Figure 3) is plotted as a function of several catchment characteristics. This distance is used as a proxy for the presence of net intercatchment groundwater flows. The black line and dots show the correlation for all stations of the Meuse basin and the colored dots (with sizes scaled to catchment area) and blue line display the catchments of the Semois River only. Left: distance to the energy limit as a function of the percentage highly fissured aquifers including karstified rocks based on the International Hydrogeological Map of Europe (IHME), indicating larger net losses as the percentage of highly fissured aquifers increases because of lower (or negative) values of the distance to the energy limit. Middle: distance to the energy limit as a function of percentage of hillslopes defined as slopes above 13% and representative for the competition between surface and subsurface drainage. Right: distance to the energy limit as a function of catchment areas of the main tributaries (up to 4000 km$^2$).





**Figure 6.** Performance indicators during the calibration (2007-2011, right column) and the validation period (2012-2016, left column) for the zero, constant and preferential models (plotted from left to right) for the Semois at Sainte-Marie, the Vierre at Straimont, the Semois at Tintigny, the Semois at Chiny and the Semois at Membre-Pont. Including net intercatchment groundwater flows leads to an improved performance in the catchment of Sainte-Marie but not in the other catchments of the Semois.



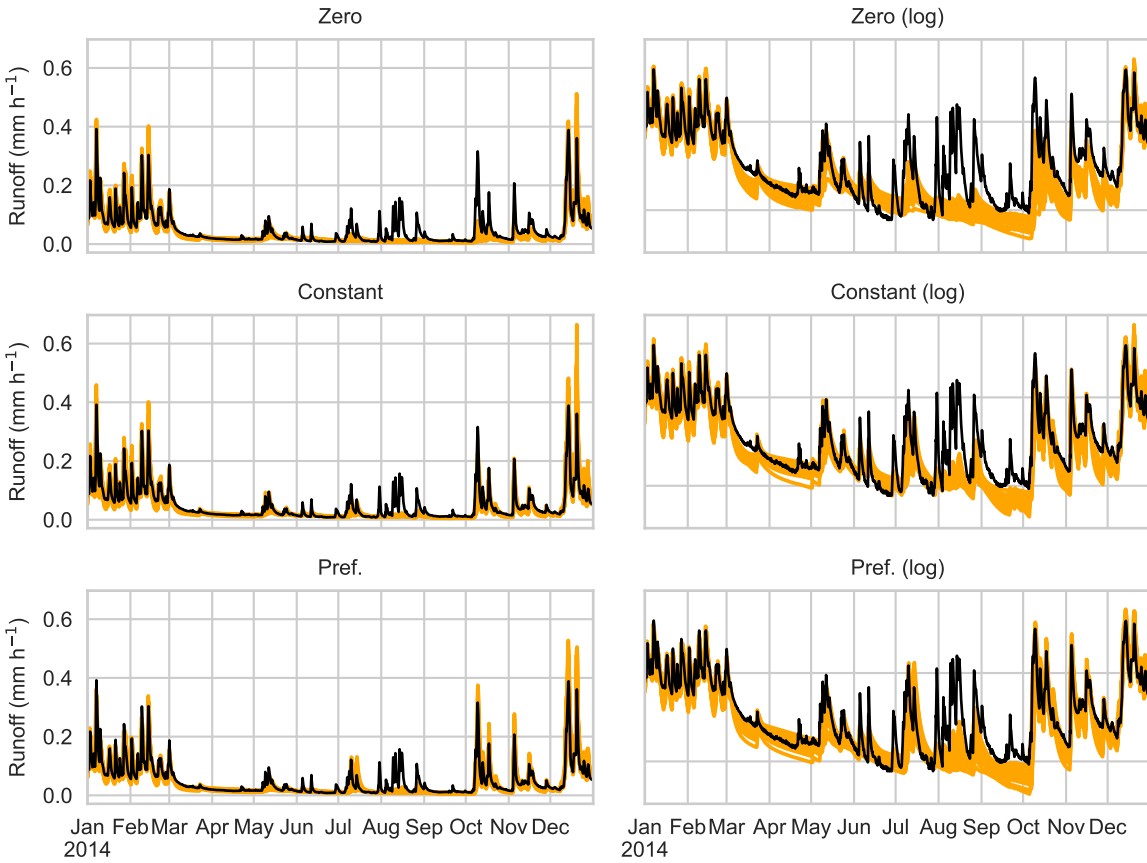

**Figure 7.** Observed (black line) and feasible realizations of modeled hydrographs (orange) in the catchment of the Semois at Sainte-Marie in 2014 for the three models (upper row: zero, middle row: constant and lower row: preferential model) on a normal (left column) and log (right column) scale. Including net intercatchment groundwater flows leads to lower simulated winter runoff (Jan-Mar) and higher runoff in the wetting up period (Oct-Nov).

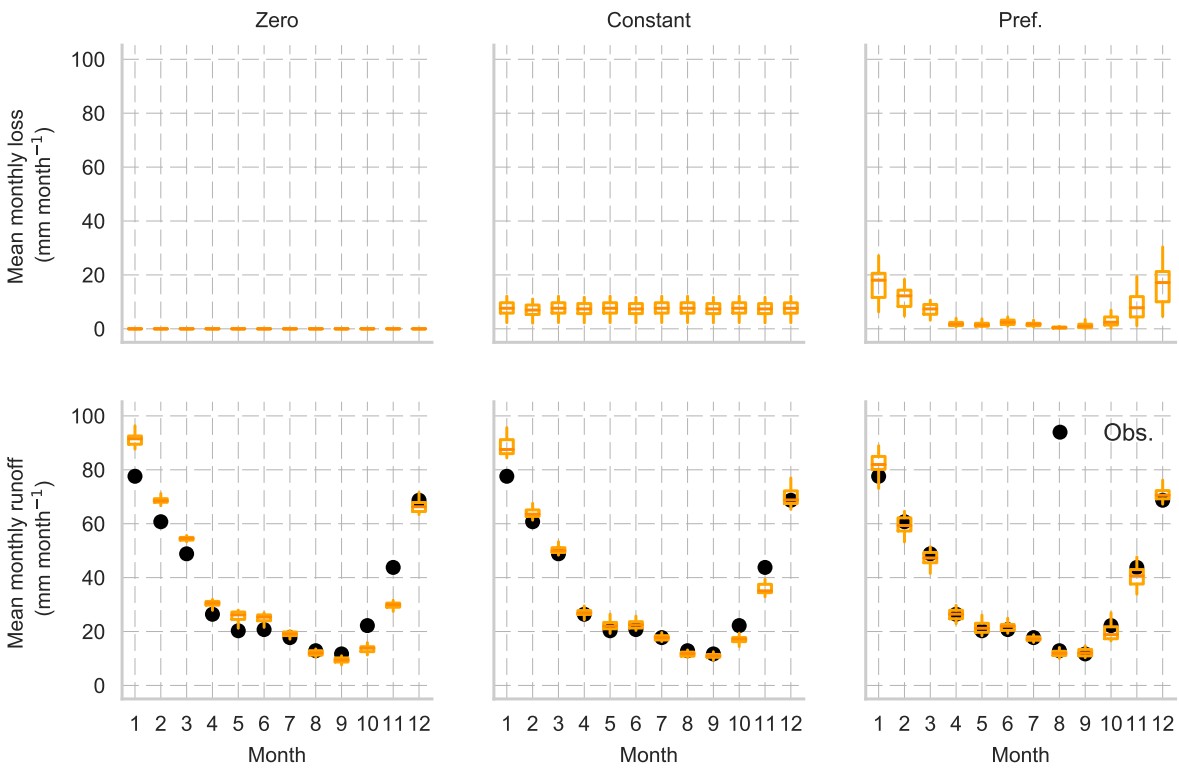

**Figure 8.** Upper row: mean monthly loss between 2007-2016 for the feasible model realizations in the Semois catchment at Sainte-Marie. Lower row: mean monthly discharge between 2007-2016 for the feasible model realizations (orange) and observations (black dots) for the three models at Sainte-Marie. The preferential model leads to better performances with lower simulated runoff in the first half year and higher runoff in the wetting up period (Oct-Nov).





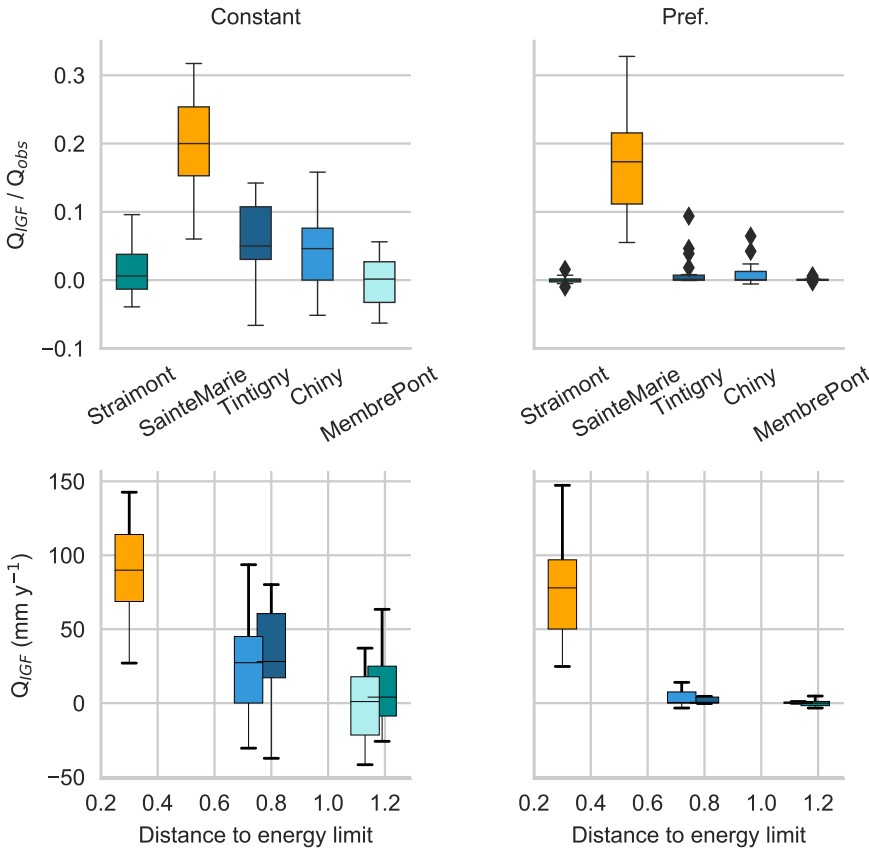

**Figure 9.** Upper row: ratio of modeled net loss over observed discharge for the Semois stations for the period 2007-2016 (positive values indicate a net loss, whereas negative values indicate a net gain) for the constant model (left) and the preferential model (right). Lower row: mean annual net intercatchment groundwater flow rates (for the feasible model realizations) as a function of the observed distance to the energy limit for the catchments of the Semois river (same color code as upper row) for the constant model (left) and preferential model (right). Both models show a decrease in net intercatchment groundwater flows as the distance to the energy limit increases.





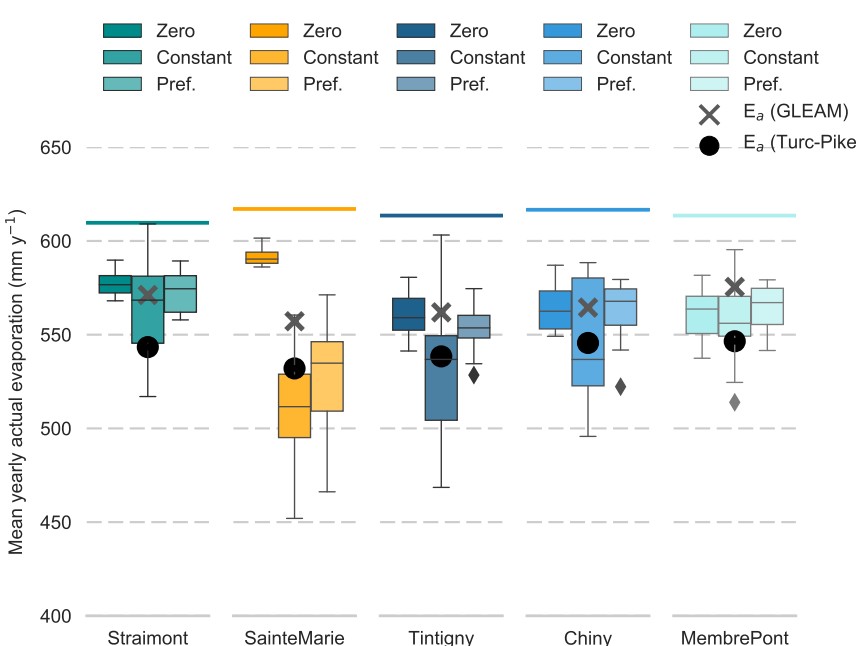

**Figure 10.** Best realizations of modeled mean annual actual evaporation in all stations in the Semois catchment for the three models during 2007-2016 (boxplots show from left to right: zero, constant and preferential model results). Colored horizontal lines indicate mean annual potential evaporation used as forcing. Estimates of actual evaporation from the Turc-Pike line are shown as black dots and GLEAM estimates are shown as grey crosses. In the catchment of Sainte-Marie, the reference model without net intercatchment groundwater flows overestimates actual evaporation compared to the other two models and Turc-Pike estimates.

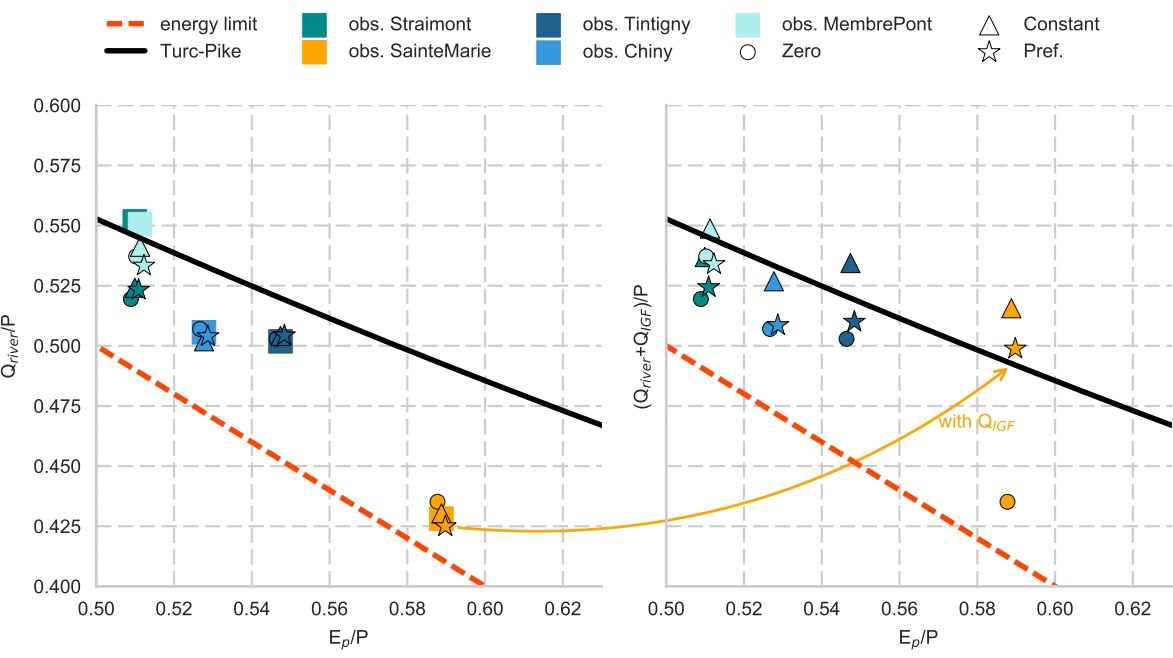

**Figure 11.** Left: dimensionless representation of $Q_{river}/P$ as a function of $E_p/P$. Long term observed values between 2007-2016 are shown together with modeled river flows (runoff from fast and slow reservoirs) using the three models for all stations of the Semois River. Right: dimensionless representation of $(Q_{river}+Q_{IGF})/P$ as a function of $E_p/P$. In this plot, we acknowledge that part of the groundwater bypasses the gauging station and consider this flow in addition to the river flow. For the catchment of Sainte-Marie, we see a shift towards the Turc-Pike line when net intercatchment groundwater flows are acknowledged.




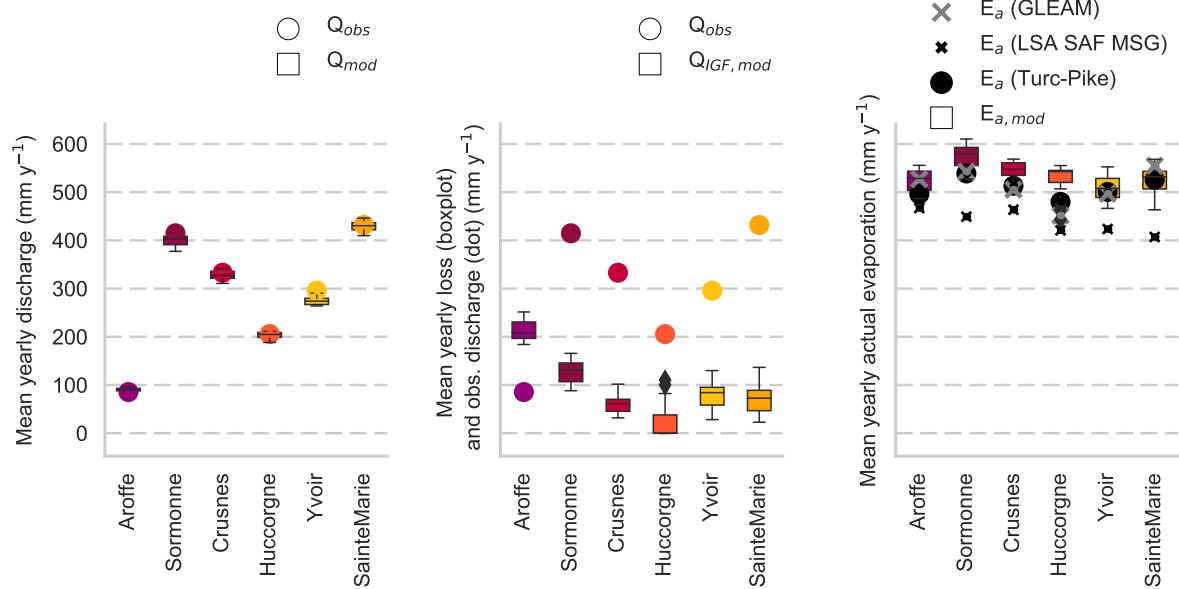

**Figure 12.** Water balance components of additional modeled catchments over the period 2007-2016. Left: modeled (boxplot) and observed (dot) mean yearly discharge overlap well. Middle: modeled mean annual net loss (boxplot) and observed mean yearly discharge (dot), showing the large proportion of net intercatchment groundwater flows especially in the Aroffe catchment. Right: modeled actual evaporation (boxplot), GLEAM actual evaporation (grey cross), LSA SAF actual evaporation (black cross) and Turc-Pike estimates (black dot). It should be noted that LSA SAF estimates are only available during the validation period (2012-2016). Model results overlap relatively well with GLEAM and Turc Pike estimates of actual evaporation, but LSA SAF estimates are lower.



**Table 1.** Catchment characteristics ($^*$Fissured denotes the percentage of highly productive fissured aquifers based on the International Hydrogeological Map of Europe, IHME). Meteorological and hydrological data are based on data between October 2006 and September 2016.

| Station | Straimont | Ste-Marie | Tintigny | Chiny | Membre P | Huccorgne | Yvoir | Belval | Pierrepont | V-le-C |
|---|---|---|---|---|---|---|---|---|---|---|
| River | Vierre | Semois | Semois | Semois | Semois | Mehaigne | Bocq | Sormonne | Crusnes | Aroffe |
| Area (km$^2$) | 182 | 143 | 381 | 738 | 1226 | 305 | 230 | 369 | 207 | 198 |
| Mean elev. (m) | 440 | 366 | 405 | 407 | 390 | 158 | 268 | 254 | 340 | 367 |
| Mean slope (-) | 0.067 | 0.044 | 0.055 | 0.060 | 0.083 | 0.026 | 0.064 | 0.066 | 0.054 | 0.060 |
| Forest (%) | 34 | 38 | 50 | 47 | 56 | 3 | 16 | 28 | 23 | 48 |
| Pasture (%) | 29 | 26 | 22 | 24 | 18 | 1.7 | 14 | 48 | 13 | 20 |
| Urban (%) | 8 | 11 | 6 | 6 | 5 | 15 | 10 | 4 | 4 | 1 |
| Crop (%) | 29 | 26 | 22 | 22 | 21 | 80 | 60 | 20 | 60 | 30 |
| Hillslopes (%) | 7.4 | 1.5 | 4.6 | 6.0 | 15 | 0.9 | 7.5 | 9.1 | 6.6 | 8.4 |
| Fissured$^*$ (%) | 0 | 63 | 27 | 16 | 9 | 16 | 71 | 48 | 94 | 72 |
| $P$ (mm y$^{-1}$) | 1176 | 1041 | 1110 | 1152 | 1183 | 753 | 867 | 1114 | 939 | 833 |
| $Q_{obs}$ (mm y$^{-1}$) | 665 | 455 | 570 | 600 | 665 | 206 | 297 | 422 | 337 | 88 |
| $E_p$ (mm y$^{-1}$) | 608 | 615 | 611 | 614 | 611 | 627 | 618 | 620 | 618 | 621 |
| $Q_{obs}/P$ (-) | 0.57 | 0.44 | 0.51 | 0.52 | 0.56 | 0.27 | 0.34 | 0.38 | 0.36 | 0.11 |
| $E_p/P$ (-) | 0.52 | 0.59 | 0.55 | 0.53 | 0.52 | 0.83 | 0.71 | 0.56 | 0.66 | 0.75 |