# Peer review of "Redressing the balance: quantifying net intercatchment groundwater flows"

_Hydrology and Earth System Sciences, 2018_

## Referee Comment (RC1) · Anonymous Referee #1 · 12 Aug 2018

"General Comments":

This paper covers a very timely topic and would be a nice addition to HESS. The concept of quantification of Inter-catchment Groundwater Flow (IGF) is still in its infancy, but its relevance to the modeling and process understanding regarding water quality and quantity is obvious. The study summarized in this paper applies a three step approach to quantify IGF that relies on the (1) comparison and analysis of observed water balance data within the Budyko framework, (2) applying a suite of different conceptual hydrological models and (3) remote sensing based estimates of actual evaporation. Their analyses suggest that IGF varies annually, and at the scale of the headwaters, IGF can make up a relatively large proportion of the water balance. At the same time, as detailed in the comments below, I do have some substantial concerns. After these

issues are resolved, I believe this paper will make a nice and impactful contribution to HESS.

"Specific comments"

Introduction

1) The introduction falls short in acknowledging recent research on the quantification of IGF. Gleeson and Manning [2008], Welch and Allen [2012] and Ameli et al. [2018] used physically-based approaches to explicitly quantify IGF. These works also explored factors controlling the IGF. It might also be useful to cite some previous works which used Budyko framework to estimate watershed-scale groundwater recharge/discharge or IGF.

2) As it is in the introduction now, the importance of the understanding of IGF is limited to improving conceptual models. In addition to that, IGF impacts (1) water quality in the higher-order streams (2) the fate and biogeochemical alteration of non-point source agricultural pollution (3) the water replenishment in economically important aquifers within arid and semi-arid mountainous regions (4) the generation and migration of petroleum and mineral deposits, and (5) the ecological functioning of the watershed. These points have been discussed in Ameli et al. [2018].

3) The current introduction did not clearly state how the current paper goes beyond the status quo and why we have to use the proposed approach to quantify IGF. As stated above, recent works explicitly quantified IGF using sophisticated physically-based hydrological models. In my opinion, the advantage of the proposed approach in this paper is to use a simple framework and widely available observations to estimate IGF. While previous approaches used extensive tracer and hydrometric observations, which are rarely available in most landscapes, to explicitly quantify IGF.

Limitations and Advances

It is good that the author explained some of the limitations of the proposed framework.
[Figure]

However, I think this part still should be extended to provide the readers with a better understanding of the applicability and limitation of the proposed framework.

1) Although the proposed framework worked well in the Muse basin with high percentage of steep hillslopes, it ignores surface storage of water in lakes and wetlands. Surface storage of water is an important element of water budget in flat lake/wetland-dominated watersheds. Water retains in these storages for decades without reaching the stream. Ignoring this element when using the proposed approach can lead to a wrong estimation of actual evaporation and IGF.

2) As the authors acknowledged, the Budyko framework is subject to uncertainties in the data used to calculate long term averages of precipitation, discharge and potential evaporation. In addition, this paper used data from different sources at different watersheds. These uncertainties limit the ability of the framework to compare the estimated IGF between watersheds. This should be clarified in this section. Having said that, the comparison made in figure 9 (lower panel) might not be robust given the different sources of data in different watersheds used in the Budyko analysis. Off course that part of the comparison made using the conceptual model is valid.

3) Similarly, the proposed framework has limited ability to estimate IGF for different scenarios of land use and climate change. IGF is a slow process with transit time of over hundreds of years (cf [Ameli et al., 2018]), and is not rapidly sensitive to most environmental changes. So it takes long time that the changes in climate and land use impact the amount of IGF (but the Budyko framework may suggest in a different manner as Q/P changes).

4) Also please clarify that the Budyko framework is only able to estimate long-term IGF and not annual IGF.

"Minor comments"

P2-L15. Delete extra period.

[Figure]

P2L33. It is true for some but not all types of solutes. Ameli et al. [2017] compared the degree to which the residence time and concentration of different solutes are corresponded.

P3L1. Gleeson and Manning [2008] used water budget analyses to calculate the actual rates of intercatchment groundwater exchanges

P3L6. Provide examples of these models and their citations

P5L2. Perhaps this last sentence could come earlier in the paragraph

P6L24. Explain the Turc-Pike framework and its assumptions

P10L24. But previous research showed different conclusions (see Ameli et al. [2018] and Gleeson and Manning [2008]). As the watershed slope increases, the water table depth increases on average, leading to more regional GW and thus more intercatchment GF.

P13L19 Use annually in the entire paper and figure labels/captions

P14L33. This is too general statement. This value may be significantly larger or smaller for different types of geological settings and watershed slope.

Reference Cited

Ameli, A. A., C. P. Gabrielli, U. Morgenstern, and J. McDonnell (2018), Groundwater subsidy from headwaters to their parent water watershed: A combined field-modeling approach, Water Resources Research, 54.

Ameli, A. A., K. Beven, M. Erlandsson, I. Creed, J. McDonnell, and K. Bishop (2017), Primary weathering rates, water transit times and concentration-discharge relations: A theoretical analysis for the critical zone, Water Resources Research, 52.

Gleeson, T., and A. H. Manning (2008), Regional groundwater flow in mountainous terrain: Threeâ̆Řdimensional simulations of topographic and hydrogeologic controls,

Water Resources Research, 44(10).

Welch, L., and D. Allen (2012), Consistency of groundwater flow patterns in mountainous topography: Implications for valley bottom water replenishment and for defining groundwater flow boundaries, Water Resources Research, 48(5).

---

## Author Comment (AC1) · 29 Aug 2018

Dear reviewer,

Thank you for the positive and constructive review of our manuscript. We value the comments and suggestions you have made to improve the manuscript and would like to respond to them below.

*"General Comments":*

*This paper covers a very timely topic and would be a nice addition to HESS. The concept of quantification of Inter-catchment Groundwater Flow (IGF) is still in its infancy, but its relevance to the modeling and process understanding regarding water quality and quantity is obvious. The study summarized in this paper applies a three step ap-*

*proach to quantify IGF that relies on the (1) comparison and analysis of observed water balance data within the Budyko framework, (2) applying a suite of different conceptual hydrological models and (3) remote sensing based estimates of actual evaporation. Their analyses suggest that IGF varies annually, and at the scale of the headwaters, IGF can make up a relatively large proportion of the water balance. At the same time, as detailed in the comments below, I do have some substantial concerns. After these issues are resolved, I believe this paper will make a nice and impactful contribution to HESS.*

*"Specific comments"*

*Introduction*

*1) The introduction falls short in acknowledging recent research on the quantification of IGF. Gleeson and Manning [2008], Welch and Allen [2012] and Ameli et al. [2018] used physically-based approaches to explicitly quantify IGF. These works also explored factors controlling the IGF. It might also be useful to cite some previous works which used Budyko framework to estimate watershed-scale groundwater recharge/discharge or IGF.*

Thank you for pointing out these recent studies on the quantification and controls of IGF. We will acknowledge them in the reviewed version of the manuscript.

*2) As it is in the introduction now, the importance of the understanding of IGF is limited to improving conceptual models. In addition to that, IGF impacts (1) water quality in the higher-order streams (2) the fate and biogeochemical alteration of non-point source agricultural pollution (3) the water replenishment in economically important aquifers within arid and semi-arid mountainous regions (4) the generation and migration of petroleum and mineral deposits, and (5) the ecological functioning of the watershed. These points have been discussed in Ameli et al. [2018].*

Thank you for this interesting point, indeed the importance of understanding IGF is not

limited to improving conceptual models, we will make sure to state this in the revised version of the manuscript.

*3) The current introduction did not clearly state how the current paper goes beyond the status quo and why we have to use the proposed approach to quantify IGF. As stated above, recent works explicitly quantified IGF using sophisticated physically-based hydrological models. In my opinion, the advantage of the proposed approach in this paper is to use a simple framework and widely available observations to estimate IGF. While previous approaches used extensive tracer and hydrometric observations, which are rarely available in most landscapes, to explicitly quantify IGF.*

Thank you for raising this issue, we agree that this paper provides a simple framework which uses widely available observations to estimate IGF and we will make sure to add this clearly in the introduction of the revised manuscript.

*Limitations and Advances*

*It is good that the author explained some of the limitations of the proposed framework. However, I think this part still should be extended to provide the readers with a better understanding of the applicability and limitation of the proposed framework.*

*1) Although the proposed framework worked well in the Muse basin with high percentage of steep hillslopes, it ignores surface storage of water in lakes and wetlands. Surface storage of water is an important element of water budget in flat lake/wetlanddominated watersheds. Water retains in these storages for decades without reaching the stream. Ignoring this element when using the proposed approach can lead to a wrong estimation of actual evaporation and IGF.*

We thank the referee for raising this interesting point. We agree that the proposed approach can lead to wrong estimations of actual evaporation and IGF in lake/wetland dominated watersheds and we will make sure to state this in the revised version of the manuscript.

[Figure]

*2) As the authors acknowledged, the Budyko framework is subject to uncertainties in the data used to calculate long term averages of precipitation, discharge and potential evaporation. In addition, this paper used data from different sources at different watersheds. These uncertainties limit the ability of the framework to compare the estimated IGF between watersheds. This should be clarified in this section. Having said that, the comparison made in figure 9 (lower panel) might not be robust given the different sources of data in different watersheds used in the Budyko analysis. Off course that part of the comparison made using the conceptual model is valid.*

If we understand this issue correctly, we should clarify that there are also uncertainties from the fact that precipitation and discharge observations are from different sources (French sources for the French part of the catchment and Belgian sources for the Belgian part). In spite of these differences, we believe that the quality of precipitation and discharge observations is sufficiently high to enable a comparison of estimated IGF between watersheds. The analysis made in Figure 9 only involves watersheds which make use of data provided by the Service Public de Wallonie and we therefore think that the comparison is robust, even in the lower panel of the plot.

*3) Similarly, the proposed framework has limited ability to estimate IGF for different scenarios of land use and climate change. IGF is a slow process with transit time of over hundreds of years (cf [Ameli et al., 2018]), and is not rapidly sensitive to most environmental changes. So it takes long time that the changes in climate and land use impact the amount of IGF (but the Budyko framework may suggest in a different manner as Q/P changes).*

Thank you for raising this point, indeed, Ameli et al. (2018) state that regional groundwater flows can have mean transit time of hundreds of years, however the distinction should be made between the mean transit time through the catchment and the mean response time of the catchment. The mean transit time characterizes the hundreds of years a water particle may need to travel from the surface where it arrives as a raindrop to the catchment outlet through deep subsurface flow paths. This process is driven by

the advective velocity of a particle. On the other side, rainfall events initiate the propagation of pressure waves through the system and enable the catchment to release water with a much faster response time. This process is driven by the celerity of the propagation of the pressure wave. The very long mean transit time of water molecules and the rapid rainfall-runoff response time imply that very old water can be released by the catchment in weeks, days or hours. As we are interested in the fast response of the propagation wave through the catchment, we believe that the framework should still be applicable to assess the impact of future land use and climate change scenarios on IGF. We will make sure to address this in the revised version.

*4) Also please clarify that the Budyko framework is only able to estimate long-term IGF and not annual IGF.*

We will clarify this in the revised manuscript.

*"Minor comments"*

*P2-L15. Delete extra period.*

Thank you for seeing this, we will delete the extra period.

*P2L33. It is true for some but not all types of solutes. Ameli et al. [2017] compared the degree to which the residence time and concentration of different solutes are corresponded.*

Thank you for raising this point, we will be more specific in the revised version of the manuscript.

*P3L1. Gleeson and Manning [2008] used water budget analyses to calculate the actual rates of intercatchment groundwater exchanges*

Thank you for pointing this out, we will include this reference in the revised version.

*P3L6. Provide examples of these models and their citations*

We will add examples and citations of these models in the revised manuscript.

*P5L2. Perhaps this last sentence could come earlier in the paragraph*

We will change this in the revised manuscript.

*P6L24. Explain the Turc-Pike framework and its assumptions*

We will add this in the revised manuscript.

*P10L24. But previous research showed different conclusions (see Ameli et al. [2018] and Gleeson and Manning [2008]). As the watershed slope increases, the water table depth increases on average, leading to more regional GW and thus more intercatchment GF.*

Thank you for raising this point, however, in their study, Gleeson and Manning [2008] assume a homogeneous subsurface because their objective is to explore the general behavior of groundwater flows on a regional scale rather than to study specific groundwater flows in a particular geological setting. In the Meuse basin, the studied flatter catchments are mainly underlain by high-permeability (potentially karstified) geological features which might be a stronger control than the watershed slope.

*P13L19 Use annually in the entire paper and figure labels/captions*

We will make sure 'annually' is used consistently throughout the paper.

*P14L33. This is too general statement. This value may be significantly larger or smaller for different types of geological settings and watershed slope.*

Thank you for raising this point, we agree that different types of geological settings and watershed slope may be more important controls than the size of the watershed, however, we show that evidence for IGF is largest in small catchments and less pronounced in larger downstream catchments, even though there are also small catchments with little evidence for it. We will make sure to make the statement more specific in the revised version.

*Reference Cited*

*Ameli, A. A., C. P. Gabrielli, U. Morgenstern, and J. McDonnell (2018), Groundwater subsidy from headwaters to their parent water watershed: A combined field-modeling approach, Water Resources Research, 54.*

*Ameli, A. A., K. Beven, M. Erlandsson, I. Creed, J. McDonnell, and K. Bishop (2017), Primary weathering rates, water transit times and concentration-discharge relations: A theoretical analysis for the critical zone, Water Resources Research, 52.*

*Gleeson, T., and A. H. Manning (2008), Regional groundwater flow in mountainous terrain: Three-dimensional simulations of topographic and hydrogeologic controls Water Resources Research, 44(10).*

*Welch, L., and D. Allen (2012), Consistency of groundwater flow patterns in mountainous topography: Implications for valley bottom water replenishment and for defining groundwater flow boundaries, Water Resources Research, 48(5).*

---

## Referee Comment (RC2) · Anonymous Referee #2 · 15 Sep 2018

Synthesis

1. This is a very interesting paper on the difficult question of an unobservable element of catchment water balance, Intercatchment Groundwater Flows (IGF). Because they are unobservable, they must be deduced from water balance anomalies, and their estimates are for this reason extremely uncertain. I recommend the publication of this manuscript after minor corrections.

Comments

2. Concerning the Introduction : You should perhaps mention that a difficulty inherent in the study of the IGF lies in the fact that conceptual models have several options to adjust the water balance, and that unfortunately they cannot afford to calibrate at the

same time a parameter for 1) IGF, 2) Precip correction, 3) PE correction... thus they have to make an assumption on the main source of error and historically modellers have had a tendency to favour precipitation correction.

3. You use both "watershed" and "catchment". Is it on purpose? If not, I would recommend simplifying the vocabulary, choosing e.g. "catchment".

4. P. 3 L.3-14 : your discussion reminds me of our own discussion of the same topic, in a paper that you may not be aware of (Mouelhi et al., 2006). There was a section poetically entitled "Is the underground water exchange parameter a fudge factor?". We showed through a proof by contradiction that IGFs "cannot be ignored on the grounds that [they are] difficult to model. The hypothesis that [IGFs] are negligible must be demonstrated by the fact that the models where it is not included are more efficient than the others since they are not overburdened by a useless additional component." We concluded that IGFs, "far from being a negligible flux of water, are an important feature of water balance modelling."

5. P6 Eq 1 and 2 : I would find it easier to follow if you used a different notation for the instantaneous flux and the integrated value (may be p and P).

6. P6 L24 : about the "Turc-Pike" formula : even if the literature sometimes use this name for the formula presented in Eq. 3, this is not fair, because the formula has been proposed almost simultaneously in France and in the Soviet Union respectively by Turc (1954) and Mezentsev (1955) : there is a detailed hydro-historical account in Lebecherel et al. (2013). I really think it should be named after both Turc & Mezentsev (and not after Turc and Pike). In our 2007 paper, we used this denomination because we had not yet rediscovered the work of Mezentsev at that time. Also, the use of the second parameter (alpha) in Eq. 3 is not standard (it's Le Moine's own modification of the formula), and since you use alpha = 1, I suggest that you stick with the classical Turc-Mezentsev formula (with only one free parameter). Last, to help your readers, you could perhaps replace gamma with n, it is more common in the literature.

References

Le Moine, N., Andréassian, V., Perrin, C., and Michel, C. 2007. How can rainfall-runoff models handle intercatchment groundwater flows? Theoretical study based on 1040 French catchments, Water Resources Research, 43.

Mezentsev, V., 1955. Back to the computation of total evaporation. Meteorologia i Gidrologia, 5: 24-26.

Mouelhi, S., Michel, C., Perrin, C. & Andréassian, V. 2006. Stepwise development of a two-parameter monthly water balance model. J. Hydrol. 318(1-4), 200-214.

Turc, L. 1954. Le bilan d'eau des sols: relation entre les précipitations, l'évaporation et l'écoulement, Ann. Agron., Série A, 5, 491–595.

---

## Author Comment (AC2) · 21 Sep 2018

Dear referee,

Thank you for your positive synthesis and detailed comments. We will take them into account to improve the manuscript and would like to shortly respond to them below.

*Synthesis*

*1. This is a very interesting paper on the difficult question of an unobservable element of catchment water balance, Intercatchment Groundwater Flows (IGF). Because they are unobservable, they must be deduced from water balance anomalies, and their estimates are for this reason extremely uncertain. I recommend the publication of this manuscript after minor corrections.*

*Comments*

*2. Concerning the Introduction : You should perhaps mention that a difficulty inherent in the study of the IGF lies in the fact that conceptual models have several options to adjust the water balance, and that unfortunately they cannot afford to calibrate at the same time a parameter for 1) IGF, 2) Precip correction, 3) PE correction. . . thus they have to make an assumption on the main source of error and historically modellers have had a tendency to favour precipitation correction.*

We agree that conceptual models have several options to adjust the water balance and even though historically modellers have had the tendency to favour precipitation corrections, this may indeed not lead to more realistic representations of the underlying processes. We will make sure to discuss this in greater detail in the introduction.

*3. You use both "watershed" and "catchment". Is it on purpose? If not, I would recommend simplifying the vocabulary, choosing e.g. "catchment".*

We indeed did not imply a distinction between "watershed" and "catchment" and we will make sure to consistently use "catchment" in the revised version of the manuscript.

*4. P. 3 L.3-14 : your discussion reminds me of our own discussion of the same topic, in a paper that you may not be aware of (Mouelhi et al., 2006). There was a section poetically entitled "Is the underground water exchange parameter a fudge factor?". We showed through a proof by contradiction that IGFs "cannot be ignored on the grounds that [they are] difficult to model. The hypothesis that [IGFs] are negligible must be demonstrated by the fact that the models where it is not included are more efficient than the others since they are not overburdened by a useless additional component." We concluded that IGFs, "far from being a negligible flux of water, are an important feature of water balance modelling."*

Thank you for referring us to the interesting paper by Mouelhi et al. (2006), which we will discuss in the revised version of the manuscript. We agree with the drawn

conclusions that IGF should be explicitly considered as they can represent an important feature of water balance modelling.

*5. P6 Eq 1 and 2 : I would find it easier to follow if you used a different notation for the instantaneous flux and the integrated value (may be p and P).*

Thank you for this suggestion, we agree that making a distinction in the symbols used for the instantaneous and integrated fluxes would make it easier for the reader and we will therefore adapt this in the revised version.

*6. P6 L24 : about the "Turc-Pike" formula : even if the literature sometimes use this name for the formula presented in Eq. 3, this is not fair, because the formula has been proposed almost simultaneously in France and in the Soviet Union respectively by Turc (1954) and Mezentsev (1955) : there is a detailed hydro-historical account in Lebecherel et al. (2013). I really think it should be named after both Turc Mezentsev (and not after Turc and Pike). In our 2007 paper, we used this denomination because we had not yet rediscovered the work of Mezentsev at that time. Also, the use of the second parameter (alpha) in Eq. 3 is not standard (it's Le Moine's own modification of the formula), and since you use alpha = 1, I suggest that you stick with the classical Turc-Mezentsev formula (with only one free parameter). Last, to help your readers, you could perhaps replace gamma with n, it is more common in the literature.*

Thank you for this interesting point concerning the historical origin of the water balance formula. We agree that it would be unfair not to acknowledge Mezentsev (1955) and we will adapt this in the revised version of the manuscript by referring to the formula as the Turc-Mezentsev formula. We also value your suggestion to adapt the formula presented in Eq. 3 to its classical form with one free parameter n, as also used in Lebecherel et al. (2013).

*References*

*Lebecherel, L., Andréassian, V., Perrin, C. 2013. On regionalizing the Turc-Mezentsev*

[Figure]

water balance formula. *Water Resources Research, 49(11), 7508-7517.*

*Le Moine, N., Andréassian, V., Perrin, C., and Michel, C. 2007. How can rainfall-runoff models handle intercatchment groundwater flows? Theoretical study based on 1040 French catchments, Water Resources Research, 43.*

*Mezentsev, V., 1955. Back to the computation of total evaporation. Meteorologia i Gidrologia, 5: 24-26.*

*Mouelhi, S., Michel, C., Perrin, C. Andréassian, V. 2006. Stepwise development of a two-parameter monthly water balance model. J. Hydrol. 318(1-4), 200-214.*

*Turc, L. 1954. Le bilan d'eau des sols: relation entre les précipitations, l'évaporation et l'écoulement, Ann. Agron., Série A, 5, 491–595.*